# On the Evolution of Stress and Microstructure in Radio Frequency-Sputtered Lead-Free (Ba,Ca)(Zr,Ti)O$_3$ Thin Films

**Runar Plünnecke Dahl-Hansen \*, Marit Synnøve Sæverud Stange, Tor Olav Sunde, Johan Henrik Ræder** 
**and Per Martin Rørvik**

SINTEF, Forskningsveien 1, 0373 Oslo, Norway; marit.stange@sintef.no (M.S.S.S.);
tor.olav.sunde@sintef.no (T.O.S.); per.martin.rorvik@sintef.no (P.M.R.)
\* Correspondence: runard@sintef.no; Tel.: +47-926-54821

**Abstract:** Thin-film piezoelectrics are widely investigated for actuators and energy harvesters, but there are few alternatives to toxic lead zirconate titanate. Biocompatible Ca- and Zr-modified BaTiO$_3$ (BCZT) is one of the most promising lead-free alternatives due to its high piezoelectric response. However, the dielectric/piezoelectric properties and structural integrity of BCZT films, which are crucial for their applications, are strongly influenced by the substrate upon which the film is grown and the related processing methods. Here, the in-plane stress, microstructure, dielectric, and piezoelectric properties of 100–500 nm thick high-temperature RF-sputtered BCZT films on industrially relevant Si-based substrates were investigated. Obtaining polycrystalline piezoelectric films required deposition temperatures $\geq$ 700 °C, but this induced tensile stresses of over 1500 MPa, which caused cracking in all films thicker than 200 nm. This degraded the dielectric, piezoelectric, and ferroelectric properties of films with larger electrode areas for applications. Films on SrTiO$_3$, on the other hand, had a compressive residual stress, with fewer defects and no cracks. The grain size and surface roughness increased with increasing deposition temperature. These findings highlight the challenges in processing BCZT films and their crucial role in advancing lead-free piezoelectric technologies for actual device applications.

**Keywords:** lead-free piezoelectrics; thin films; RF sputtering; residual stress; structural integrity; energy harvesting; microelectromechanical systems (MEMS)

## 1. Introduction

Lead-containing ferroelectric perovskite oxides, such as Pb(Zr, Ti)O$_3$ (PZT) and (PbMg$_{1/3}$Nb$_{2/3}$O$_3$)$_{1-x}$-(PbTiO$_3$)$_x$ (PMN-PT), possess some of the largest piezoelectric responses, which are required for sensors and actuators based on thin-film piezoelectric microelectromechanical systems (piezoMEMSs) [1,2]. However, their content of toxic lead raises environmental concerns [3] and renders them inapplicable for a range of biomedical applications, such as bone regenerative therapy [4] and in-body energy harvesters for powering implanted medical devices [5,6]. The lead content of today's piezoelectrics is a recurring topic of discussion and it is widely recognized that as soon as lead-free materials display piezoelectric properties comparable or exceeding those of e.g., PZT, lead-containing materials will be phased out for this application [7]. Thus, multiple lead-free materials are currently being investigated. Among the candidates, biocompatible and non-cytotoxic BaTiO$_3$-based compositions, particularly Ba$_{0.85}$Ca$_{0.15}$Zr$_{0.1}$Ti$_{0.9}$O$_3$ (BCZT), are promising as they display exceptionally high piezoelectric and dielectric compliance [4,8,9]. In fact, charge coefficients ($d_{33}$) exceeding 250 pm/V for thin-film actuators [10] and figures of merit (FOMs) for piezoelectric energy harvesting (PEH) exceeding that of PZT [9] have been reported. But to introduce BCZT thin films into piezoMEMS-based sensors and actuators it is essential to prepare defect-free films with low residual stress, controlled microstructure, and large electrode areas.

Applying thin films for piezoMEMS relies on maximizing the tolerance of the piezoelectric layer stack to field or stress-generated elastic deformation, without electromechanically breaking down. This is well illustrated for PEH, in which the average power output ($P_{rms}$) is proportional to the piezoelectric stress coefficient ($e_{31,f}$), the electrode area ($A$), the film thickness ($t_f$), and the generated strain squared ($\epsilon_i^2$): $P_{rms} \propto e_{31,f} A t_f \epsilon_i^2$ [11]. It is clear that maximizing the piezoelectric response, electrode-active areas, and mechanical deformation in thick, robust, and defect-free piezoelectric thin film layer stacks is imperative for high-performing PEHs. Although BCZT thin films have been realized using chemical [12,13] and physical [14–16] deposition methods, processing μm thick films remains a major challenge as they display a high propensity for cracking when mechanically stressed [17–19]. Because thin films are restricted by the substrate, they are often subject to a considerable amount of residual stress ($\sigma_R$) after processing. Residual and applied stress affect the film's mechanical, dielectric, piezoelectric, and ferroelectric properties [20] and must be closely monitored and controlled, especially if high crystallization temperatures are required and the fracture toughness is low, as is often the case for $BaTiO_3$-derived compositions [21].

At the thin-film level, it is generally desirable for piezoelectrics to have large, well-coalesced, and well-oriented grains with few grain boundaries, as this enhances the ferroelectric and dielectric responses [12,22]. The substrate as well as the choice of the seed-layer material influence grain size, orientation, and film texture. Platinized silicon is perhaps the most industrially relevant platform for thin-film piezoMEMS development today. A thin layer of (001)-oriented $LaNiO_3$ (LNO) is commonly added as a seed oxide electrode to promote (001)-orientation and the piezoelectric properties of ferroelectric perovskite oxides, to act as a chemical diffusion barrier, and to prevent fatigue [16,17,23]. However, if the substrate is much thicker than the film, the mechanical properties and stress state of the piezoelectric layer stack system are substrate-dominated. Si and $SiO_2$ have relatively low CTEs, $\alpha_{Si} \sim 3.6$ μm/K, and $\alpha_{SiO_2} \sim 0.65$ μm/K, respectively [24], whereas for $BaTiO_3$-derived compositions, the CTE is comparatively high. BCZT-based compositions have reported values of $11 < \alpha_{BCZT} < 17$ μm/K, and required crystallization temperatures of up to 800 °C [25,26]. In comparison, PZT has a CTE of $\alpha_{PZT} \sim 4.5$ μm/K, and is typically deposited around 650 °C [2,20,27]. Thus, better CTE-matched substrates, such as (001)-oriented $SrTiO_3$ (STO) with $\alpha_{STO} \sim 9.4$ μm/K [15], or MgO with $\alpha_{MgO} \sim 13.5$ μm/K [28], are often used experimentally. STO with an in-plane lattice constant about 2% smaller than BCZT ($a_{STO} \sim 3.91$ Å vs. $a_{BTO} \sim 4.00$ Å) has also been shown to promote epitaxial film growth with compressive strain and favored c-axis orientation. Furthermore, making STO electrically conducting by Nb-doping to function as a bottom electrode with high temperature stability is attractive for thin film integration. For this reason, STO has become one of the more studied templates for BCZT thin films [16,19]. These substrates are, however, not industrially viable. Thus, direct integration of BCZT thin films with Si is still being actively pursued in the literature, but stress measurements of BCZT thin films, which are important to understand the mechanical and piezoelectric properties, are scarce.

In this work, we investigate the in-plane residual stress of high-temperature RF-sputtered BCZT thin films by curvature measurements using white-light interferometry, and correlate the in-plane stress to the microstructure evolution and piezoelectric properties of BCZT thin films. Four selected substrates are compared: oxidized Si, platinized Si, $LaNiO_3$ (LNO)-coated platinized Si, and Nb-doped $SrTiO_3$. The first three substrates are chosen to evaluate the possibility of direct integration with industrially relevant thin-film piezoMEMS platforms for sensors and actuators. While growth on substrates without a bottom electrode is relevant for actuators and PEHs with interdigitated top electrode designs, studying the properties of thin film on different bottom electrode interfaces is of relevance for applications where the field is applied across the layer stack. Commonly used Nb:STO [15] is chosen to compare the substrate-induced properties of the BCZT thin films. The evolution of in-plane stress is evaluated using Stoney's approximation. Thermal and growth contributions to the stress are assessed by varying the deposition temperature and

deposition pressure, respectively. Microstructural features and out-of-plane piezoelectric and ferroelectric properties are measured and discussed.

## 2. Materials and Methods

Figure 1a–d schematically show the substrates investigated in this work: (a) $SiO_2$/Si (Si hereafter), (b) Pt/$TiO_2$/$SiO_2$/Si (Pt hereafter), (c) LNO/Pt/$TiO_2$/$SiO_2$/Si (LNO/Pt hereafter), and (d) $SrTiO_3$ doped with 0.5 wt% Nb (Biotain Crystal, Xiamen, China) (STO hereafter). All (001)-oriented 150 mm Si wafers, Figure 1a–c, were thermally oxidized to an oxide thickness of $t_{SiO_2}$ = 500 nm. For the substrates in Figure 1b,c, a 30 nm Ti adhesion layer and a 100 nm (111) Pt bottom electrode were sputtered at 450 °C (Memsstar Endura cluster, Applied Materials, Livingston, UK). The substrate in Figure 1c has an additional 30 nm thick (001)-oriented $LaNiO_3$ oxide electrode seed layer, deposited at 580 °C by pulsed laser deposition (Solmates Piezoflare, Enschede, The Netherlands), acting as an orientation promoter for BCZT. All Si-based wafers were laser-diced into substrates of 50 × 50 mm$^2$ before BCZT sputtering, whereas the (100)-oriented 10 × 10 × 0.5 mm$^3$ STO substrates were used as received. The Nb-doping of STO ensures an electrically conducting substrate for simple contacting. BCZT films of 100–500 nm ± 1% thickness (measured using a J. A. Woolam a-SE spectroscopic ellipsometer) were RF-sputtered (Flextura 200, Polyteknik, Østervrå, Denmark) from a 4″ diameter hot-pressed $Ba_{0.85}Ca_{0.15}Zr_{0.1}Ti_{0.9}O_3$ target (Kurt J. Lesker, Hastings, UK). All substrates were cleaned in ultrasonicated acetone, isopropanol, and DI-water and then Ar plasma-etched with a power of 50 W at $5 × 10^{-2}$ mbar for 5 min prior to film deposition. The target power density and Ar:$O_2$ ratio were kept constant at 3 W/cm$^2$ and 9:1 for all depositions. After ignition burst and a 1 min target precleaning, the deposition pressure, $p_D$, was flow-controlled to obtain a stable plasma at $4 × 10^{-3}$–$1.6 × 10^{-2}$ mbar. Two main deposition series were investigated: (i) changing $T_D$ at a constant $p_D = 8 × 10^{-3}$ mbar, and (ii) changing $p_D$ at a constant $T_D = 700$ °C. The temperature was ramped up to 200 °C $\leq T_D \leq$ 800 °C, and then down to room temperature at a rate of 10 K/min, in vacuum with no post-deposition annealing. For selected films, an Au (200 nm)/Ti (10 nm) top electrode/adhesion layer was deposited by electron beam deposition (Polyteknik Tornado, Østervrå, Denmark) and structured into 100 × 100 μm$^2$ pads by photolithography and etching in potassium iodide (Au-etch) and $H_2O$:$H_2O_2$:50% HF (20:1:1, Ti-etch). These were used for the measurement of ferroelectric/dielectric properties using a TF2000 analyzer (AixACCT Gmbh, Aachen, Germany). Bipolar polarization vs. electrical field measurements (*P-E*) were conducted at 10 Hz and $V_{P,max}$ = 400 kV/cm, bipolar capacitance and dielectric loss vs. electrical field measurements (*C-E*) with a small-signal amplitude of 200 mV, on a potentiostatic offset-voltage of up to $V_{P,max}$ = 400 kV/cm, and a small-signal frequency of 1 kHz.

Atomic force microscopy (AFM) and piezoresponse force microscopy (PFM) (both Bruker Multimode 8, Bruker, Billerica, MA, USA) with an Pt/Ir-coated Sb-doped Si tip in contact mode were used to characterize the topography and the piezoelectric response, respectively. The roughness ($R_q$) was calculated from the AFM measurements using the root mean square of heights:

$$R_q = \sqrt{\frac{1}{n}\sum_{i=1}^{n}\left(Z_i - \overline{Z}\right)^2} \qquad (1)$$

Here, $Z_i$ is the vertical height of grain $i$, and $\overline{Z}$ is the average grain height. The initial roughness values of the substrates used were $R_{q,Si} < 0.1$ nm, $R_{q,Pt} = 10 \pm 1$ nm, $R_{q,LNO/Pt} = 10 \pm 3$ nm, and $R_{q,STO} = < 0.1$ nm, for Si, Pt, LNO/Pt, and STO, respectively. Grain sizes (horizontal extent) were extracted from neighboring grain boundaries. Phase purity, film orientation, and crystallinity were examined by X-ray diffraction (XRD, Bruker AXS D8 Discovery, Billerica, MA, USA) and transmission electron microscopy (TEM, FEI Titan G2 60-300, Hillsboro, OR, USA) for microstructural insight.

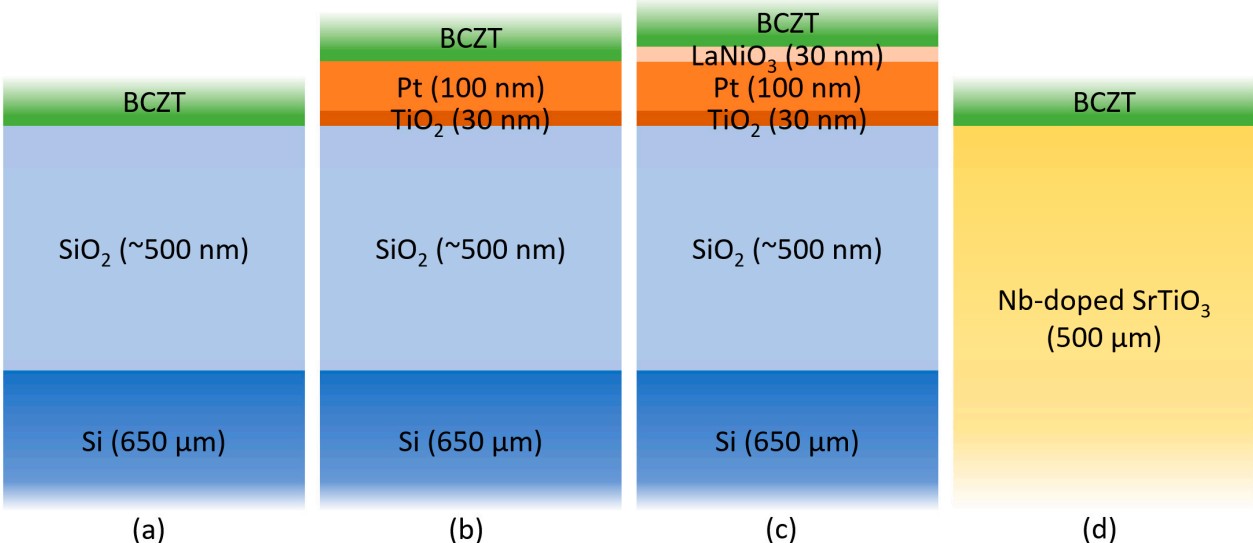

**Figure 1.** (**a**–**d**) Schematic of substrates investigated in this work: (**a**) Si, (**b**) Pt, (**c**) LNO/Pt, and (**d**) STO.

For stress considerations, thicknesses of Si and STO substrates of $t_{Si}$ = 650 μm and $t_{STO}$ = 500 μm, i.e., much thicker than the thickness of the deposited BCZT thin films, $t_f$, were chosen as experimental boundary conditions. Thus, Stoney's approximation could be used to calculate the in-plane film stress ($\sigma$) from the measured substrate bending before and after deposition [29]:

$$\sigma = \frac{E_s t_s^2}{6 t_f (1 - \nu_s)} \left( \frac{1}{r} - \frac{1}{r_0} \right) \tag{2}$$

Here, $E_s$ is the Youngs modulus of the substrate, $t_s$ is the substrate thickness, $\nu_s$ is the Poisson ratio of the substrate, $r$ is the radius of curvature of the film–substrate system after deposition, and $r_0$ is the radius of curvature of the substrate before deposition. Here, we used $E_{Si}$ = 170 GPa, $\nu_{Si}$ = 0.28 [29] and $E_{STO}$ = 274.8 GPa and $\nu_{STO}$ = 0.24 [30]. White-light interferometry (WYKO NT9800, Oak Ridge, TN, USA) was used to measure the macroscopic 3D z(x, y) topography before and after film deposition. The substrate curvatures from measurements were fitted to a semicircle to obtain $r$ and $r_0$. Table 1 summarizes the measured thicknesses and residual stresses, $\sigma_R$, in all films in the layer stacks before depositing BCZT. This was used for assessing the validity of the measurements by modelling the resulting thermal stress and substrate deformation using the classical laminated plate theory. Only crack-free films, obtained with a 190 ± 12 nm thickness, measured using a J. A. Woolam a-SE spectroscopic ellipsometer, were used for measuring the BCZT stress using Equation (2).

**Table 1.** Thickness and residual stress for the materials in the substrates.

| Material | Measured Thickness [μm] | Measured Built-in Stress [MPa] |
|---|---|---|
| LaNiO$_3$ | 0.03 | 370 ± 10 |
| Pt | 0.1 | 670 ± 50 |
| TiO$_2$ | 0.05 | 445 ± 21 |
| SiO$_2$ | 0.5 | −300 ± 30 |
| Si | 650 | ~0 |

## 3. Results

### 3.1. Morphological and Structural Properties

Figure 2 shows the TEM cross-section images of a 125 nm thick BCZT film deposited on LNO/Pt at $T_D = 700$ °C. The films grew into fan-shaped polycrystalline columnar structures. The X-ray diffractograms of BCZT films deposited on Si at $400 < T_D < 800$ °C (Figure 3) show only Si substrate reflections at 400 °C and 600 °C, while at 800 °C the reflections at ~39° and ~83° correspond to (111) and (222) perovskite reflections, respectively. This confirms that the BZCT film on Si grew preferentially in the (111) direction. Corresponding XRD data for BCZT films deposited onto Pt and LNO/Pt substrates were dominated by the Pt (111) reflection at 39.9° masking the BCZT (111) reflection, and in addition showed (101)/(110) and (001/100) BCZT reflections (Figure S1a). For STO, BCZT reflections indicate mainly (001/100) orientation (Figure S1b).

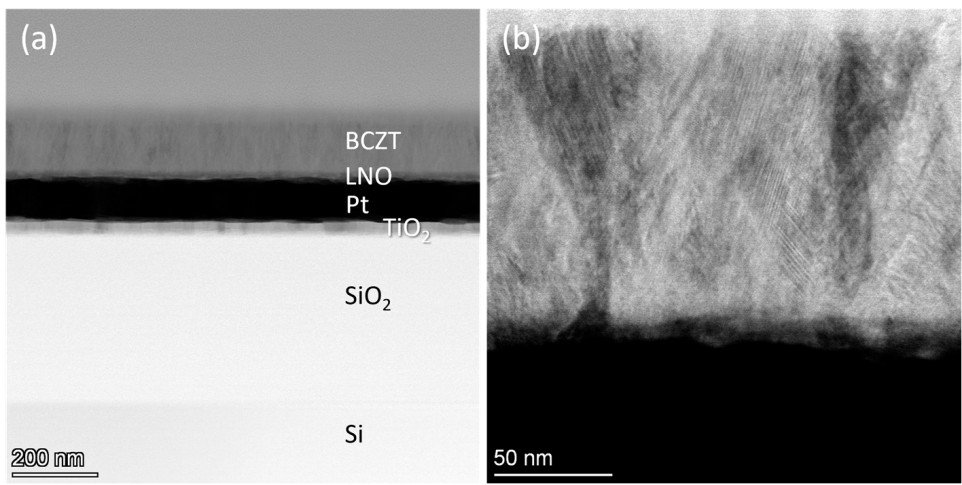

**Figure 2.** TEM images of a cross-section of BCZT/LNO/Pt/Si stack with BCZT deposited at 700 °C: (**a**) bright-field TEM image with layers indicated, and (**b**) larger magnification bright-field TEM image of the BCZT film (gray).

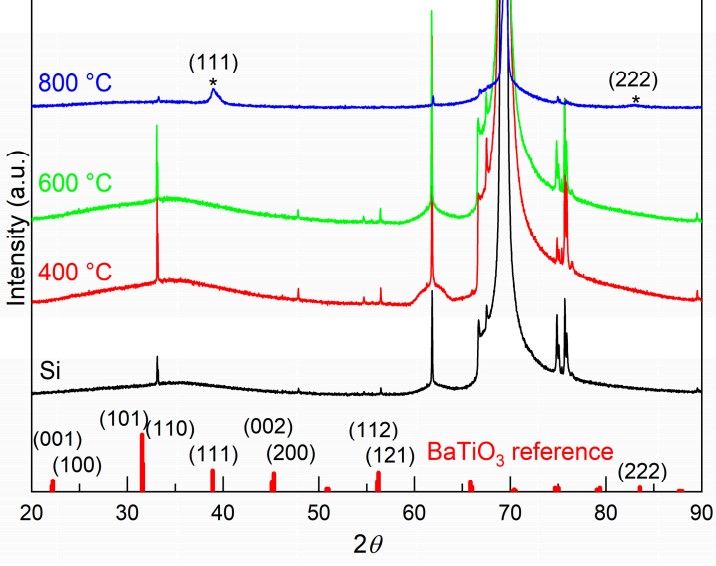

**Figure 3.** X-ray diffractograms of the Si substrate and ~100 nm BCZT thin films deposited at 400 °C, 600 °C, and 800 °C onto Si. Asterisks indicate BCZT reflections. The sticks at the bottom refer to tetragonal BaTiO$_3$ (COD 2100858).

Figure 4a exemplifies a $1 \times 1$ μm² AFM image series of BCZT with $t_f > 200$ nm deposited at 400–800 °C on Pt/Si substrates. Table 2 summarizes the measured $R_q$ and grain size. Only amorphous BCZT films were obtained below $T_D = 500$ °C with $R_q$ consistently less than 1 nm, regardless of substrate type and film thickness. At $T_D = 600$ °C, partial crystallization occurred, resulting in a mix of smooth and rough areas, some containing crystalline grains and others without. X-ray diffraction indicates polycrystalline BCZT films for $T_D \geq 700$ °C (Figure 3 and Figure S1). Increasing $T_D$ above 700 °C was associated with a significant increase in $R_q$ and grain size, exceeding 11 and 70 nm, respectively, at $T_{D,max} = 800$ °C.

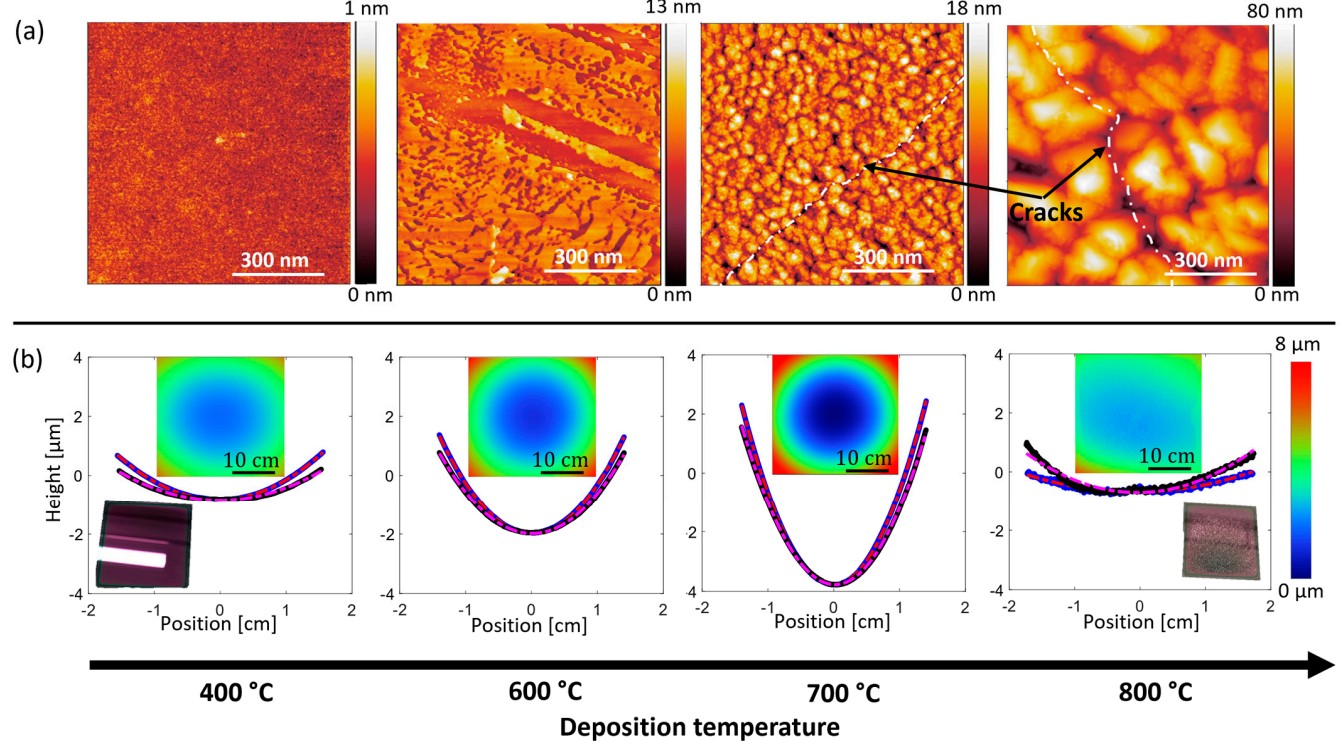

**Figure 4.** (**a**) AFM images showing the microstructural evolution of as-deposited BCZT films of $t_f > 200$ nm on platinized Si with increasing $T_D$. Amorphous films were obtained at 400 °C, partly crystallized films at 600 °C, and fully crystallized films at 700 °C. A considerable increase in grain size, roughness, and extension of cracks is observed for the film deposited at 800 °C. (**b**) Curvature evolution as a function of $T_D$ for $t_f > 200$ nm BCZT films on platinized Si. The inset shows 2D x/y scans with colored height profiles, from which the line curvature is extracted.

**Table 2.** Microstructure and in-plane stress results for BCZT film thicknesses below 200 nm for different deposition temperatures at deposition pressure of $8 \times 10^{-3}$ mbar.

| Deposition Temperature, $T_D$ [°C] | Surface Roughness, $R_q$ [nm] | Average Grain Size [nm] | Modelled Stress [MPa] | Measured in-Plane Stress, $\sigma_R$ [MPa] | | |
|---|---|---|---|---|---|---|
| | | | | **BCZT/Si** | **BCZT/Pt/Si** | **BCZT/LNO/Pt/Si** |
| 200 | $0.2 \pm 0.1$ | N/A | 210 | $-168 \pm 83$ | $-431$ | $-512$ |
| 400 | $0.3 \pm 0.1$ | N/A | 581 | $342 \pm 123$ | $550 \pm 123$ | $531 \pm 91$ |
| 600 | $1.2 \pm 0.5$ | $3 \pm 1$ | 1085 | $725 \pm 115$ | $787 \pm 81$ | $839 \pm 42$ |
| 700 | $2.2 \pm 0.8$ | $8 \pm 2$ | 1526 | $1383 \pm 92$ | $1553 \pm 97$ | $1564 \pm 68$ |
| 800 | $11.5 \pm 2.3$ | $70 \pm 10$ | 1721 | $1971 \pm 190$ | $2153 \pm 216$ | $2188 \pm 210$ |

### 3.2. Film Stress

Figure 4b shows the curvature evolution for a series of BCZT films with $t_f > 200$ nm deposited on Pt substrates at $400 < T_D < 800$ °C. The inset pictograms show the substrate at $T_D = 400$ °C and $T_D = 800$ °C. Before deposition, the investigated substrates had initial curvatures of $r_{0,Si} = 1817 \pm 1123$ m, $r_{0,Pt} = 147 \pm 24$ m, and $r_{0,LNO/Pt} = 83 \pm 6$ m. At $T_D = 200$ °C, the radius of curvature of the film–substrate system after deposition ($r$) was larger than the radius of curvature before deposition ($r_0$), and thereby $\sigma_R < 0$ was true for all substrates, indicating the films were under compressive stress. For all films deposited on Si-based substrates at $T_D \geq 400$ °C, the radius of curvature was smaller after deposition (in the range of 15–100 m, Table S1), giving $\sigma_R > 0$, indicating tensile-stressed films. Although heavily cracked at 800 °C, resulting in stress alleviation and substrate-flattening, areas with adherent film allowed for AFM images to be taken. It is, however, noted that all films of $t_f > 200$ nm deposited at $T_D \geq 700$ °C on Si-based substrates displayed large cracks and were disregarded for further analysis. In comparison, the deposition of BCZT onto STO at 700 °C gave compressively stressed films of around 2 GPa (see Table S1). This is rationalized from the larger unit cell of BCZT compared to STO.

Figure 5a shows the measured in-plane stress as a function of $T_D$ for Si-based substrates with $t_f < 200$ nm, i.e., BCZT films with few or no extended cracks. With relatively small variations across the substrates, the results depict a general trend: Starting as compressive at $T_D = 200$ °C, $\sigma_R$ becomes tensile at $T_D = 400$ °C. $\sigma_R$ increases approximately linearly between 600 and 800 °C, with a slope of 9 MPa/°C, exceeding 2000 MPa at 800 °C for all substrates. The values for thermal stresses measured here are in reasonable agreement with values reported elsewhere [31]. However, the modelling results, indicated by the red line, show relatively large discrepancies between measured and modelled stress. Table 3 and Figure 5b summarize the measured $\sigma_R$ for the different deposition pressures, $p_D$, at a constant $T_D = 700$ °C. $\sigma_R$ increases with $p_D$ at an approximate rate of 90 MPa/µbar. It is notable that $\sigma_R$ exceeds 1 GPa for all depositions exceeding $T_D = 600$ °C, regardless of $p_D$. We note that experiments using room-temperature deposition followed by rapid thermal annealing (RTP) at temperatures from 700 °C to 900 °C were carried out on $t_f < 200$ nm to emulate the thermal contribution to the total stress, as shown in Figure S2. Though an overall reduction in $\sigma_R$ was found, in the range $\sigma \sim 1400$–2300 MPa, fully crystallized films could not be obtained at 700 °C. The morphological trend of the post-annealed films with temperature was similar to the high-temperature-deposited films.

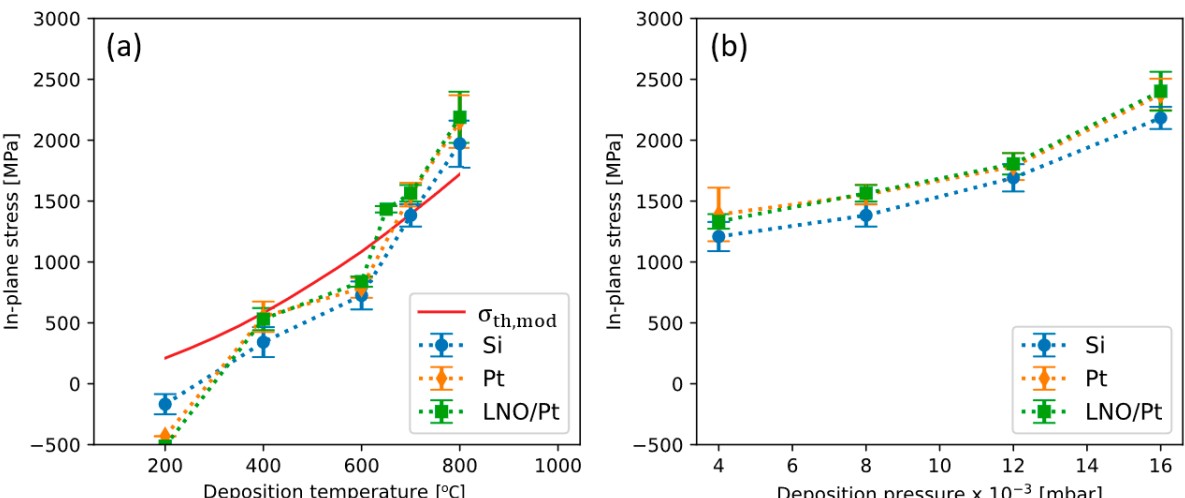

**Figure 5.** (**a**) Measured in-plane stress as a function of deposition temperature, $T_D$, with constant deposition pressure of $=8 \times 10^{-3}$ mbar; predicted in-plane stress using the laminate theory model is shown by the solid red line. (**b**) Measured in-plane stress as a function of $p_D$ at $T_D = 700$ °C.

**Table 3.** In-plane stress for different deposition pressures at deposition temperature of 700 °C. All measurements are conducted on BCZT films with $t_f \leq 200$ nm.

| Substrate | In-Plane Stress [MPa] | | | |
|---|---|---|---|---|
| | $4 \times 10^{-3}$ mbar | $8 \times 10^{-3}$ mbar | $12 \times 10^{-3}$ mbar | $16 \times 10^{-3}$ mbar |
| BCZT/Si | $1209 \pm 120$ | $1383 \pm 92$ | $1691 \pm 112$ | $2183 \pm 91$ |
| BCZT/Pt/Si | $1390 \pm 220$ | $1553 \pm 81$ | $1784 \pm 111$ | $2376 \pm 129$ |
| BCZT/LNO/Pt/Si | $1333 \pm 60$ | $1564 \pm 68$ | $1806 \pm 88$ | $2402 \pm 160$ |

*3.3. Electrical Properties*

Figure 6 shows microscope images of the BCZT/electrode (Au, $t_{f,Au} \sim 200$ nm) for films with $t_f > 200$ nm deposited at 700 °C on STO (a) and LNO/Pt-coated Si (b). We chose 700 °C to minimize the number of defects, and residual thermal stresses while still having a polycrystalline film. It is clear that while appearing continuous and defect-free on STO, relatively large cracks and pinholes penetrate both the BCZT film and top Au-electrode on the LNO/Pt substrates. The corresponding capacitance field (*C-E*), polarization field (*P-E*) and PFM measurements in Figure 7 for the substrates in question, reflect this observation: on STO, the capacitance is low, yet a ferroelectric loop appears. The capacitance is the highest on LNO/Pt, and a significantly inflated *P-E* loop indicates large leakage. The latter is most pronounced for BCZT/Pt, indicating that the film is more or less completely short-circuited. This is also in line with dielectric losses ($\tan(\delta)$, not shown here) of ~0.7, ~0.15, and ~0.25 for the STO, LNO/Pt, and Pt substrates, respectively.

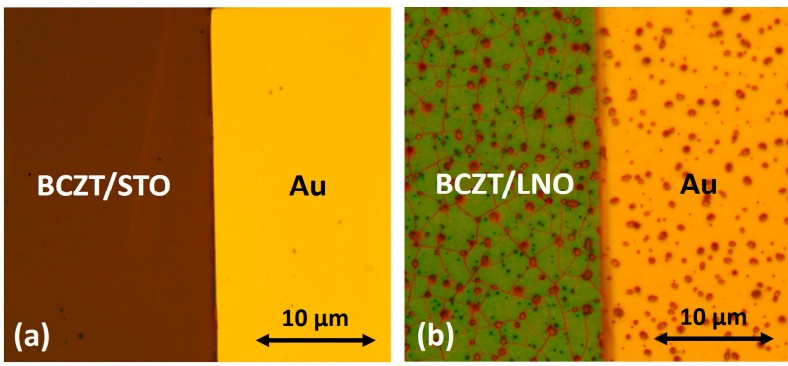

**Figure 6.** Optical microscope images showing the electrode edge and films on $t_f > 200$ nm thick BCZT films deposited at 700 °C on (**a**) STO and (**b**) LNO/Pt-coated Si.

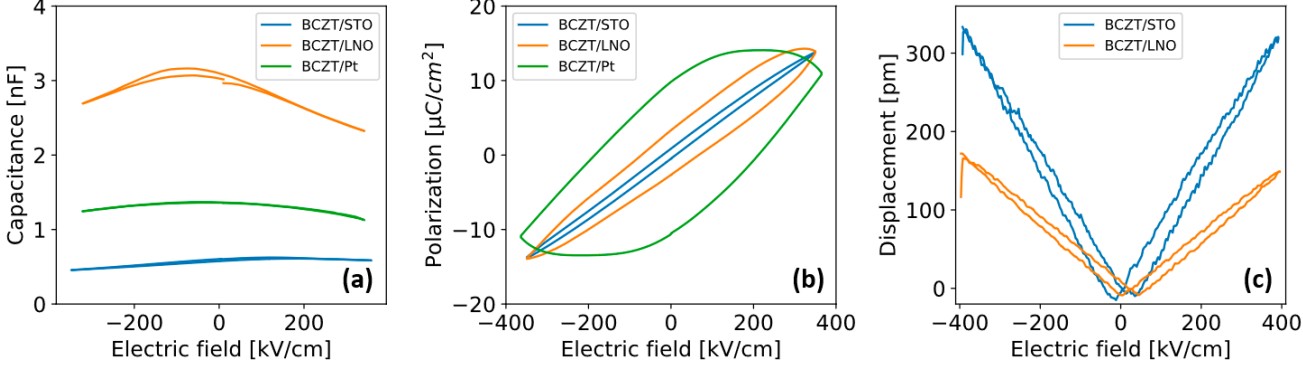

**Figure 7.** Capacitance field (**a**), polarization field (**b**), and piezoelectric force microscopy (PFM) (**c**) measurements of BCZT films with $t_f > 200$ nm grown at 700 °C on the three substrates: STO (blue), LNO/Pt (orange), and Pt (green). Piezoelectric responses could only be detected for BCZT on STO and BCZT on LNO/Pt due to the high defect density for thicker films grown on Pt.

PFM measurements for $t_f > 200$ nm, shown in Figure 7c, could only be successfully conducted on STO and LNO/Pt. BCZT deposited on STO displays the highest piezoelectric response with an out-of-plane charge coefficient of $d_{33,f} \sim 36$ pm/V. On LNO/Pt, the charge coefficient is $d_{33,f} \sim 17$ pm/V, with the piezoelectric loop shifted towards higher voltages.

## 4. Discussion

The residual stress ($\sigma_R$) in thin films is related to processing, and depends on the film and substrate materials, layer stack design, and deposition technique used. For polycrystalline films deposited using energetic physical vapor deposition (PVD) methods, such as radiofrequency (RF) magnetron sputtering, the major contributions to $\sigma_R$ are thermal ($\sigma_{th}$) and growth ($\sigma_{gr}$) stresses [27,32]. $\sigma_{th}$ develops due to a mismatch in the coefficient of thermal expansion (CTE, $\alpha$) of the film and the substrate, upon cooling from the deposition temperature ($T_D$) to room temperature. Growth stress, $\sigma_{gr}$, arises as a result of nucleation and growth processes on the substrate during deposition, including adatom bombardment, surface diffusion, and island coalescence. Generated microstructural flaws such as crystallographic defects, local nonstoichiometry, and inclusion of particles at grain boundaries can significantly alter the stress state of the film [32]. For piezoceramic thin films such as BaTiO$_3$-derived compositions, adatom surface diffusivity is low, and $\sigma_{gr}$ can therefore be significant.

All the crystalline BCZT films on Si-based substrates in this work are under significant tensile stresses, remaining intact only below 200 nm. The measured stresses are comparable across all Si-based substrates, which is reasonable given that Si dominates the elastic properties of the thin-film systems. It was observed that BCZT films with $t_f > 200$ nm deposited on native 6″ Si wafers remained intact post deposition up to $T_D > 700$ °C, but were found to crack easily upon further handling. Films deposited on oxidized Si measured a lower $\sigma_R$ than the other substrates, which suggests that $\sigma_R$ here is just below the films' critical stress, avoiding immediate film cracking when exposed to ambient conditions. Though not investigated in detail here, the morphology, texture, template, and layer stack design are expected to play a key role. Clearly, the thermal stress dominates the stress contribution for $T_D > 700$ °C. This was found to be true for both in situ-crystallized and post-crystallized (RTP, see Figure S2) films. However, the additional contribution from $\sigma_{gr}$ in Figure 5b demonstrates some tunability by changing the deposition pressure. At low $T_D$, the films are amorphous, and the adatom surface diffusion low, which produces a net compressive stress. This is generally expected for metal oxide films [32,33]. Although $\sigma_R$ is always tensile for partly or fully crystallized films, at $T_D > 600$ °C kinetic processes on the substrate surface during growth have a large effect on both microstructure and $\sigma_{gr}$ [32]. During island coalescence, increased adatom energy at lower pressures (less scattering in the plasma) increases particle inclusion in grain boundaries, and exerts compressive stress on the grains [34], and even more so for high-melting-point ceramics, which typically have relatively low adatom mobility. Despite this, the minimum pressure of $4–16 \times 10^{-3}$ mbar for maintaining a stable plasma in our system was not enough to reduce $\sigma_R$ below 1 GPa for the films deposited on Si. This is much too high to bring any of such films into actuator or harvester applications, i.e., the absolute film stress dictates cracking at some critical stress level [35,36], and the large built-in stress limits the applicable strain before the films will crack, altogether impeding practical applications.

There are very few reports overall on the in-plane stress of BCZT on Si in the literature. One exemption, by Biswas et al. [37], found smaller values using XRD, yet also experienced film peeling for films annealed at 800 °C, comparable to this work. This was found for BCZT films deposited directly onto oxidized Si substrates without a bottom electrode, and here these were also found to have slightly lower stress than the ones grown on platinized Si (see Figure 5).

It is noteworthy that piezoelectric properties in the literature are measured for the most part by PFM on films below 300 nm in thickness, which reflects the overall challenge related to in-plane stress for integrating BaTiO$_3$-derived thin films with Si. Generally,

thicker films have a higher propensity for cracking than thinner films, as a direct effect of the coupled criterion, an extension of the classical Griffith–Irwin criterion to explain the cracking of thin films [38]. In essence, for a material to crack, the normal tensile stresses along a potential crack path must exceed the material's tensile strength, and enough energy must be available to induce cracking. As the potential energy increases with thickness for a mechanically loaded film, cracks can be initiated at smaller loads for thicker films than for thinner films [36]. The presence of defects in a film decreases the tensile strength and increases the likelihood of initiating a crack. The cracking of even very thin films here suggests that the coupled criterion is met as a result of the considerable residual stresses. Furthermore, the low reported fracture strength (~0.9 GPa) and fracture toughness (~0.4 MPa$\sqrt{m}$) for BaTiO$_3$-derived films in general [21] increase the material's propensity for cracking. This contrasts with, for instance, PZT, whose CTE and fracture toughness (0.6–1.8 MPa$\sqrt{m}$) allow for integration with Si-based substrates [36,39]. We note that other lead-free piezoelectric materials with lower CTE may be better suited for RF magnetron sputtering of thin films onto Si substrates, for instance, K$_{0.5}$Na$_{0.5}$NbO$_3$ with a reported CTE of $7.5 \times 10^{-6}$ K$^{-1}$ between 434 °C and 790 °C, and which has been used to deposit films of ~3.0 μm by sputtering [40,41].

Strategies for reducing $\sigma_R$ to directly integrate BCZT with Si have been tried out. These include buffer layers, deposition bias substrates during deposition, post-deposition annealing to control microstructure and texture, multilayer deposition, and plasticizers for chemical-solution-deposited films [33]. Nevertheless, thick defect-free BCZT films with large electrode areas well integrated with Si remain to be repeatedly demonstrated. Although promising results have been demonstrated using, e.g., sol–gel [42] or sacrificial layers [43], having Si dominate the elastic properties of the overall film–substrate system during processing indeed imposes considerable challenges for any direct integration of technologically applicable BCZT thin films.

Microstructurally, we observe the thin films crystallizing with grain sizes ranging from about 8 to 70 nm in the present study. Increasing grain size with increasing temperature can be attributed to a higher degree of coalescence of smaller grains into larger grains, as has been reported before [12,22]. While larger grains will give larger piezoelectric responses [44], it will be at the expense of less stress alleviation at grain boundaries. This can decrease the fracture strength for films with larger grains [45], requiring other pathways for low-stress integration with Si than RF-sputtering.

Fine-grained thin films have been associated with a mix of paraelectric cubic and ferroelectric tetragonal phases. The *C-E* and *P-E* characteristics in Figure 7a,b are in line with this, and have been observed before for films below 200 nm [46]. The *C-E* curves show no switching peak (Figure 7a), while a weak signature of ferroelectricity, however, with low remnant polarization is measured for BCZT on STO in the *P-E* loop (Figure 7b). The dielectric stiffening—i.e., a decrease in small-signal capacitance with increasing offset bias field—is measured and points towards a ferroelectric nature of the film. High in-plane tensile stress can lead to a *c*-to-*a* ferroelectric domain reorientation, and more *a*-domain-dominated films [20]. Films with more *a*-domains display higher capacitances and lower piezoelectric responses than films with more *c*-domains. This trend can be extended to the present films: on STO, the capacitance is low, the $d_{33,f}$ high, and a ferroelectric response is measured that is comparable to reports in the literature [15,31,42]. This is opposite to Pt- and LNO/Pt-coated Si substrates, where the capacitance is high, $d_{33,f}$ is low, and the ferroelectric response is absent (due to high leakage). A high leak here is attributed to films and electrodes containing a high number of stress-induced cracks and defects, as illustrated in the micrograph of Figure 6b. BCZT deposited on Pt is more defective with higher dielectric losses and a more inflated *P-E* loop than BCZT deposited on LNO/Pt. This is rationalized by LNO promoting more (001)-oriented films, which have overall better dielectric and piezoelectric characteristics than, e.g., (111)-oriented films. Indeed, LNO as a buffer layer on Si substrates is known to also improve the survival rate of thicker ferroelectric films [10]. Films derived, for example, by sol–gel [12] or off-axis

RF sputtering [19] offer more promising fabrication routes for higher quality films than presented here. However, finding strategies that avoid using Si as the host substrate during actual film processing appears a necessity for bringing thicker BCZT films into actuator and harvesting applications.

## 5. Conclusions

In conclusion, 100–500 nm thick polycrystalline $Ba_{0.85}Ca_{0.15}Zr_{0.1}Ti_{0.9}O_3$ (BCZT) thin films were integrated on four different substrates using high-temperature RF magnetron sputtering: Si as an industrially viable substrate, bare or with a Pt (111) seed and bottom electrode or an LNO(100)-seed oxide bottom electrode. Films deposited on Nb-doped $SrTiO_3$ were measured for comparison. A minimum deposition temperature of 700 °C was required for obtaining polycrystalline films displaying piezoelectric and ferroelectric characteristics. Curvature measurements and Stoney's approximation showed tensile in-plane residual stresses exceeding 1500 MPa at 700 °C and 2100 MPa at 800 °C on Si-based substrates. Overall, this resulted in severe cracking of all films thicker than 200 nm with degraded dielectric, ferroelectric, and piezoelectric characteristics. Although the residual tensile stress could be reduced by decreasing the deposition pressure, films with residual stresses below 1 GPa could not be obtained on the Si substrates. This is too much for actuator and harvesting applications, questioning the common pursuit for direct integration of BCZT thin films with Si. In comparison, BCZT thin films sputtered onto $Nb:SrTiO_3$ substrates were under compressive stress, had fewer defects, and displayed better piezoelectric and ferroelectric characteristics than all films deposited on Si.

**Supplementary Materials:** The following supporting information can be downloaded at: https://www.mdpi.com/article/10.3390/act13030115/s1, Figure S1: X-ray diffractograms of (a) LNO/Pt substrate and BCZT thin film deposited at 700 °C onto LNO/Pt, and (b) STO substrate and BCZT thin film deposited at 700 °C onto STO; Figure S2: AFM images and measured stress of films deposited at room temperature and subsequently post-annealed; Table S1: Radius of curvature before and after deposition, and calculated average in-plane stress, for BCZT films deposited at 700 °C at deposition pressure $8 \times 10^{-3}$ mbar.

**Author Contributions:** Conceptualization, R.P.D.-H. and P.M.R.; methodology, R.P.D.-H., M.S.S.S., T.O.S., J.H.R. and P.M.R.; formal analysis, R.P.D.-H.; investigation, R.P.D.-H. and M.S.S.S.; writing—original draft preparation, R.P.D.-H. and P.M.R.; writing—review and editing, R.P.D.-H., M.S.S.S., T.O.S., J.H.R. and P.M.R.; project administration, J.H.R. and P.M.R.; funding acquisition, J.H.R. and P.M.R. All authors have read and agreed to the published version of the manuscript.

**Funding:** This research was funded by the Research Council of Norway, grant numbers 250184 and 194068.

**Data Availability Statement:** The data presented in this study are available on request from the corresponding author.

**Acknowledgments:** The authors gratefully acknowledge Patricia Almeida Carvalho for acquiring the TEM images, Amin Shahrestani Azar for finite element modelling of the thermal stresses, and Roman Papšík and Raúl Bermejo for assistance with layer stack modelling and valuable discussions.

**Conflicts of Interest:** The authors declare no conflicts of interest.

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
