# Peer review of "On the Evolution of Stress and Microstructure in Radio Frequency-Sputtered Lead-Free (Ba,Ca)(Zr,Ti)O3 Thin Films"

_actuators, doi:10.3390/act13030115_

Round 1

Reviewer 1 Report

Comments and Suggestions for Authors

In general, the authors should point out what is new in this paper. In the final discussion, it seems like all the results match results being reported before, which is good, but naturally gives readers the impression that nothing new is done in this work.

131: Ti does not get etched in KI. Only KI is not enough.
129: Even with no data, it would be helpful to indicate if post baking alters the stress - especially post baking at normal lithography temperatures (e.g. photoresist baking temperatures).
139: SPM includes AFM. It is better to indicate what kind of SPM is used. From the context, you are using PFM.

Fig6(a),(b): measurement conditions? Why no hysteresis in (a)? The leak current does not affect capacitance measurement? From the graph it seems that you are sweeping voltage and measuring current, where you get C = i / (dV/dt). If this is true, you should also get hysteresis if there is a leak. If this is not how you measured the capacitance, you should specify how it is measured.

298: The cracking is not well explained here. In your case, the stress is not related to the thickness; and while more strain energy is with thicker films, more energy is also needed to propagate the crack in thicker films. I think a more reasonable explanation is the Griffth criterion for brittle materials, where the defects density and the size (thickness in your case) plays a crucial role.

Author Response

Dear reviewer, 

Please find the author-responses in the attached word-file

Best regards, 

Runar Dahl-Hansen

Reviewer 2 Report

Comments and Suggestions for Authors

The manuscript describes in detail (in an extended form) the technology of depositing Ca- and Zr-modified BaTiO3 (BCZT) films on substrates with various buffers SiO2/Si, Pt/TiO2/SiO2/Si, LNO/Pt/TiO2/SiO2/Si and SrTiO3 doped with 0.5 wt% Nb in order to obtain optimal defect-free controlled microstructure with low residual stress, necessary to produce films with high piezoelectric properties. This fragment of the manuscript contains a number of valuable details about the technology and may be useful for groups working on oxide layers with piezoelectric properties with significant potential for applications in sensors and actuators. The rest of the manuscript concerns the determination of the stresses of the layers and their piezoelectric and ferroelectric properties. They vary significantly depending on the substrate temperature in the deposition process and the partial pressure of gases used in the RF-sputtering process and the type of buffer (in the case of Si substrates). It was found that in layers deposited on various Si substrates (regardless of the type of buffer used), the optimal piezoelectric properties are far from the optimal ones exhibited by BCZT thin films sputtered onto Nb:SrTiO3 substrates. Therefore, significant difficulties have been demonstrated in pursuing integration of BCZT thin films with Si substrates. I believe that the manuscript is worth publishing and may constitute a valuable source for further efforts to obtain thin-film structures with piezoelectric properties.

The reviewed manuscript is an example of work in the field of "materials science" and, in my opinion - as a physicist not involved in materials science - it meets the basic criteria in this field: We manufacture and process materials, design and construct structures using materials, analyze failures of materials striving to improve the performance of the product we are designing. This manuscript meets the above criteria. It describes the technological details necessary to produce BCZT layers, and provides the basic parameters of structure and microstructure (as well as stresses and defects) affecting properties important from the point of view of applications (piezo effect).

The main question addressed by the research is: how to improve the performance of piezoelectric BCZT films integrated with industrially available Si substrates? This is not an original problem, but it is important from the perspective of materials science. The choice of an appropriate substrate is crucial in the case of oxide films deposited using sputtering at high temperatures with subsequent heating at T<~ 700 C. Due to high processing temperatures the oxide films cannot be fully integrated onto Si wafers. For this reason, the main result of the research is (unfortunately) a negative conclusion that the integration of BCZT layers with a Si substrate does not lead to systems with optimal piezoelectric properties and "works remains to successfully integrate thick applicable BCZT films with wafer level Si for sensors and actuators". In the future, it may be worth focusing on choosing the right buffer layer or using a technology other than sputtering (e.g. sol-gel?).

Despite these negative conclusions, which are supported by careful research (supported by figures and adequate references), I believe that the manuscript may be helpful for further efforts in this field.

Author Response

(The authors gave the same response as above.)

Reviewer 3 Report

Comments and Suggestions for Authors

Well writtend and presented paper on RF-sputtered ferroelectric thin films.

No need degK, just K.

Table - space between +/- for ease of reading. Subscripts.

What is error / varaiability in thickness measurements/no. meansurement?

P-E loops are somewhat lossy but strain - field data is good. The lossy P-E loops could be discussed a little more.

Nice work.

Comments on the Quality of English Language

Generally good.

Author Response

(The authors gave the same response as above.)

Reviewer 4 Report

Comments and Suggestions for Authors

This article focuses on the study of (Ba,Ca)(Zr,Ti)O3 thin films film properties.There is a lot of literature in actuators and harverters, and therefore the author could not introduce any novelty in this manuscript. In fact, this work, based on grown piezoelectric fim method, could be the basis for an interesting article. However, I have the following comments:

1 Title need to more emphasized with this investigation. Actuatores and harverters results have not been presented. This title is a bit misleading. Please change it.

2. Introduction is too long. The novelty and the research gaps could be more emphasized at the end of the introduction. It is suggested to the author to include a discussion section with some paragraph presented in introduction section. A comparison with the state of the art must be made in a discussion section. Please focus on (Ba,Ca)(Zr,Ti)O3 thin films and its effects on structural properties.

3. PLease put different sections such as morphological, structural, stress, and electric properties.

4. Please identify the highlight of this paper. Measurement results about the piezoelectric coefficient, damping or loss of the thin film could be more convincing about the improvement of the thin film.

5. More experimental SEM results on surface quality (cross section, tilted..) need to be given and explained in detail. A deeper insight into this characterization would be very helpful.

6. The results are not presented in a concise manner, but the conclusion is too long. However some sections need to be moved and inserted elsewhere to allow for better sequencing of explanations.

7. In the final conclusion is too long and it must contain the new results from the authors own research.

8. Figure 1 is not representative, split this figure in three

9. The quality of some figure is very poor, the scale bar and all label are too small (increase the size and put them in bold).

Following your consideration of the various comments I will be pleased to reconsider an appropriately revised manuscript.

Comments on the Quality of English Language

The English language is clearly not proofread; there are syntax and spelling errors in every other paragraph. Sometimes these errors make it difficult to understand the meaning of the sentence.

Proposal of abstract corrected : 

Thin film piezoelectrics are widely investigated for actuators and energy harvesters, but there are few alternatives to toxic lead zirconate titanate. Biocompatible Ca- and Zr-modified BaTiO3 (BCZT) is one of the most promising lead-free alternatives due to its high piezoelectric response. However, the dielectric/piezoelectric properties and structural integrity of BCZT films, which are crucial for the application, are strongly influenced by  industrially relevant Si substrates and related processing methods. Here, the in-plane  stress, microstructure, dielectric, and piezoelectric properties of 100-500 nm thick RF-sputtered BCZT films on industrially relevant Si-based substrates are investigated.  In order to obtain  polycrystalline piezoelectric films, deposition temperatures above 700°C were required. However this led to tensile stresses of over 1500 MPa, which causes cracks in all films thicker than 200 nm. This degraded the dielectric, piezoelectric and ferroelectric properties of films with larger electrode-areas for applications. Films on SrTiO3, on the other hand, had, exhibited a residual compressive stress, had fewer defects and no cracks. The grain-size and surface        roughness increased with increasing deposition-temperature. These findings highlight the challenges processing  BCZT films and their crucial role in the further development of lead-free piezoelectric technologies for specific applications.

Author Response

(The authors gave the same response as above.)

Round 2

Reviewer 4 Report

Comments and Suggestions for Authors

Accept in present form